# Genetically Engineered Probiotic *Limosilactobacillus reuteri* Releasing IL-22 (LR-IL-22) Modifies the Tumor Microenvironment, Enabling Irradiation in Ovarian Cancer

**DOI:** 10.3390/cancers16030474

**Published:** 2024-01-23

**Authors:** Diala F. Hamade, Michael W. Epperly, Renee Fisher, Wen Hou, Donna Shields, Jan-Peter van Pijkeren, Brian J. Leibowitz, Lan G. Coffman, Hong Wang, M. Saiful Huq, Ziyu Huang, Claude J. Rogers, Anda M. Vlad, Joel S. Greenberger, Amitava Mukherjee

**Affiliations:** 1Department of Radiation Oncology, UPMC Hillman Cancer Center, Pittsburgh, PA 15232, USA; hamadedf3@upmc.edu (D.F.H.); epperlymw@upmc.edu (M.W.E.); fisherr3@upmc.edu (R.F.); houw@upmc.edu (W.H.); shieldsd@pitt.edu (D.S.); bjl40@pitt.edu (B.J.L.); huqs@upmc.edu (M.S.H.); greenbergerjs@upmc.edu (J.S.G.); 2Department of Food Science, University of Wisconsin-Madison, Madison, WI 53706, USA; vanpijkeren@wisc.edu; 3Department of Medicine, University of Pittsburgh, Pittsburgh, PA 15260, USA; coffmanl@upmc.edu; 4Department of Biostatistics, University of Pittsburgh, Pittsburgh, PA 15260, USA; how8@pitt.edu (H.W.); huangz6@upmc.edu (Z.H.); 5ChromoLogic, LLC, Monrovia, CA 91016, USA; crogers@chromologic.com; 6Division of Cancer Prevention, National Cancer Institute, Rockville, MD 20850, USA; anda.vlad@nih.gov

**Keywords:** genetically engineered, probiotic, *Limosilactobacillus reuteri*

## Abstract

**Simple Summary:**

Ovarian cancer is the most lethal gynecological cancer worldwide, and there is an urgent need for a cure. Previously, we established that a second-generation probiotic expressing interleukin-22 (LR-IL-22) is a radiation mitigator, and now, we have assessed its effect in an ovarian cancer mouse model. LR-IL-22 treatment improved survival, upregulated PD-L1 protein expression on cancer cells, and recruited CD8+ T cells to tumors in whole-abdomen irradiated (WAI) mice. The addition of LR-IL-22 to a therapy regimen that includes WAI, chemotherapy, and immunotherapy can facilitate a safe and effective protocol to reduce tumor burden, increase overall survival, and improve the quality of life of an ovarian cancer patient.

**Abstract:**

Despite recent advances in cancer therapy, ovarian cancer remains the most lethal gynecological cancer worldwide, making it crucial and of the utmost importance to establish novel therapeutic strategies. Adjuvant radiotherapy has been assessed historically, but its use was limited by intestinal toxicity. We recently established the role of *Limosilactobacillus reuteri* in releasing IL-22 (LR-IL-22) as an effective radiation mitigator, and we have now assessed its effect in an ovarian cancer mouse model. We hypothesized that an LR-IL-22 gavage would enable intestinal radioprotection by modifying the tumor microenvironment and, subsequently, improving overall survival in female C57BL/6MUC-1 mice with widespread abdominal syngeneic 2F8cis ovarian cancer. Herein, we report that the LR-IL-22 gavage not only improved overall survival in mice when combined with a PD-L1 inhibitor by inducing differential gene expression in irradiated stem cells but also induced PD-L1 protein expression in ovarian cancer cells and mobilized CD8+ T cells in whole abdomen irradiated mice. The addition of LR-IL-22 to a combined treatment modality with fractionated whole abdomen radiation (WAI) and systemic chemotherapy and immunotherapy regimens can facilitate a safe and effective protocol to reduce tumor burden, increase survival, and improve the quality of life of a locally advanced ovarian cancer patient.

## 1. Introduction

Despite aggressive therapy, including cytoreductive surgery and platinum-based chemotherapy, women diagnosed with ovarian cancer have a five-year overall survival rate of less than 50%, making it the most lethal gynecologic cancer worldwide. Late diagnoses account for its relatively poor prognoses as most women present with symptomatic malignant peritoneal dissemination. More than 80% of patients will have an initial complete response to current standards of care consisting of surgery and platinum/taxane chemotherapy. Unfortunately, a majority will develop recurrent disease within the first 2 years, with poor responses to subsequent treatment modalities [1]. Novel therapeutic strategies are thus urgently needed to improve survival, and irradiation can be an effective treatment modality [2,3,4]. Historically, the use of adjuvant radiation for the management of ovarian cancer has been limited by intestinal toxicity as the low dose used to meet upper abdomen organ tolerance is considerably lower than the radical dose needed to treat solid tumors, resulting in poor therapeutic efficacy. However, recent advances in radiotherapy techniques such as intensity-modulated radiotherapy (IMRT) and stereotactic body radiotherapy (SBRT), in combination with targeted therapies, may pave the way for its resurgence as a successful approach in curing ovarian cancer. Furthermore, given that the disease is limited to the abdominal cavity in most women presenting with advanced ovarian cancer, a method to mitigate radiation-induced intestinal toxicity is essential for therapeutic fractionated whole abdomen irradiation (WAI). We recently established the role of an intraoral gavage of genetically engineered probiotic *Limosilactobacillus reuteri* releasing IL-22 (LR-IL-22) as an effective radiation mitigator [5]. Here, we evaluated its role in ovarian cancer by using an immune-competent syngeneic mouse model. We hypothesized that the LR-IL-22 gavage would enable a reduction in tumor burden by modifying the tumor microenvironment and, subsequently, improve overall survival in female C57BL/6MUC-1 mice with widespread abdominal syngeneic 2F8cis ovarian cancer by reducing radiation-induced intestinal toxicity, which could potentially facilitate the addition of therapeutic WAI to novel protocols in the management of ovarian cancer. 

## 2. Materials and Methods

### 2.1. Mice and In Vivo Animal Model

Adult male and female C57BL/6 mice and C57BL/6 mice with knock-in allele-lgr5-EGFP-IRES creERTZ were housed 5 females and 4 males per cage and fed standard laboratory chow and deionized drinking water. MUC1^+/−^ mice (129X/SvJ), which are transgenic for the human mucin 1 (MUC1.Tg) antigen, have been described elsewhere [6,7]. The mice received either intraperitoneal or subcutaneous injections of syngeneic 2F8cis ovarian cancer tumor cells and were sacrificed at a specific time point or when their body weights decreased over 20%. All mice were treated according to the protocols approved by the University of Pittsburgh Institutional Animal Care and Use Committee (IACUC).

### 2.2. Ovarian Cancer Cell Lines

The 2F8 and 2F8cis (Cis-platinum resistant) ovarian cancer cell lines derived from murine orthotopic ovarian cancer tumors have been described previously [6,7]. These cell lines were obtained from Dr. Anda Vlad, Ph.D., who may be contacted at the following address: Dr. Anda Vlad, Division of Cancer Prevention, National Cancer Institute, Rockville, MD 20850, USA.

### 2.3. Irradiation

The adult mice received either a single fraction of 9.25 Gy or 12 Gy total body irradiation (TBI), 13.3 Gy partial body irradiation (PBI) with one hind limb shielded (5% of bone marrow), or a fractionated regimen of whole abdomen irradiation (WAI) at a dose of 24 Gy in four fractions (one fraction per day). TBI was performed using a JL Shepherd model 68A cesium irradiator (JL Shepherd and Associates, San Fernando, CA, USA) at 280 cGy per minute. Whole abdominal irradiation (WAI) was performed using a Varian Trilogy Linear Accelerator (Varian Medical Systems, Palo Alto, CA, USA). The radiation dosimetry and field accuracy and reproducibly were performed using previously published methods [8]. For delivery of the WAI, the mice were placed in a 3 cm × 40 cm field with only their abdomens in the irradiation field, shielding their necks, thoracic regions, and lower limbs. The mice were irradiated at 600 Mu/min using 6 MV photons at a 100 cm source-to-surface distance (SSD). For delivery of the PBI, a SmART+ small animal X-ray irradiator (Precision X-ray, Inc., Madison, CT, USA), using a copper filter, was used. Four mice were arranged around a 10 cm × 10 cm field so that the body of each mouse would be in the field, with one leg outside of the field. The leg outside of the field received a dose of 0.12 Gy. 

All radiation parameters followed previously published protocols [7,9].

### 2.4. Gavage Administration of the LR and LR-IL-22 or Intraperitoneal Delivery of the IL-22 Protein

The methods for the LR-IL-22 and *L. reuteri* (LR) production, as well as administration of the IL-22 protein by intraperitoneal (IP) injections, have been previously published [10]. Briefly, the mice were gavaged with 200 microliters containing numbers of bacteria, ranging from 10^6^ to 10^10^ of either LR or LR-IL-22 measured by optimal density. Other mice received IP injections of 20 mg/kg of IL-22 protein at the same time points as the other groups (Pepro Tech, Cranbury, NJ, USA, 21022). The mice were randomly divided into the following groups: control mice and mice treated with intraoral gavages of 10^9^ control empty bacteria LR, LR-IL-22 in 200 µL of saline, or IP injections of 20 mg/kg of IL-22 protein (0.1 mg/kg) in 100 µL of saline. For the TBI and PBI studies, the mice received their respective treatments at 24 h post-irradiation. During the fractionated WAI studies, the mice received their respective treatments at serial times. 

### 2.5. Dose–Response Curves of the Number of Gavaged L. reuteri Producing IL-22 (LR-IL-22)

The mice were irradiated to 9.25 Gy TBI, which took 3–4 min, and 24 h later, they were gavaged with 200 μL of saline solution containing LR-IL-22. All groups of mice received 200 μL of saline, which contained different numbers of bacteria (10^6^, 10^7^, 10^8^, 10^9^, and 10^10^ bacteria; n = 15 animals per group).

### 2.6. RNA-seq

The Lgr5+ GFP+ mice (at 10–12 weeks of age) were irradiated to 12 Gy total body irradiation. Twenty-four hours later, the mice were gavaged with saline, IL-22 protein (0.1 mg/kg), LR (10^9^ cells), or LR-IL-22 (10^9^ cells) in 200 uL of saline. Twenty-four hours later, (two days after irradiation), the mice were sacrificed and their small intestines were removed, made into single-cell suspensions, and analyzed on a MoFlo Astrios High Speed Sorter flow cytometer (Beckman Coulter Life Sciences, Brea, CA, USA). The Lgr5+ GFP+ cells, as well as the non-GFP+ cells, were collected and frozen in Trizol (Thermo Fisher Scientific, Waltham, MA, USA). For the irradiated mice, the single-cell suspensions from the intestines of two mice were combined into one sample to obtain enough Lgr5+GFP+ RNA for the RNAseq analysis. The cells were thawed and the RNA was extracted and frozen. The RNA was shipped to Medgenome (Foster City, CA, USA) for the RNAseq analysis. The RNA library was made using a Takara SMART-Seq v4 Ultra-low Input RNA kit on a NovaSeq PE100/150 system (Illumina, San Diego, CA, USA). For each sample, we obtained 20 million paired-end (40 million total) reads.

The oligonucleotides used for the 16S sequencing were as follows:

Universal 16S Fwd: TCGTCGGCAGCGTCAGATGTGTATAAGAGACAGCCTACGGGNGGCWGCAG; 

Universal 16S Rev: GTCTCGTGGGCTCGGAGATGTGTATAAGAGACAGGACTACHVGGGTATCTAATCC; and Universal 16S internal sequencing: TATAAGAGACAGCCTACGGG.

### 2.7. Statistical Methods

For the experiments comparing the PD-L1+ cells and the CD8+ T cells between the treatment groups (control, IR, or IR plus LR-IL-22) where the IR was TBI, PBI, or WBI, the data were summarized in each subgroup using means ± standard deviations (SD). At each day of measurement, we compared each treated group with the control. Also, for each treatment, we compared each day back to day 0. All comparisons were completed based on a two-way ANOVA model, followed by a *t*-test using the ESTIMATE statement in Proc GLM of SAS 9.4 (SAS Institute Inc., Cary, NC, USA). In this ANOVA model, the treatment, the day, and their interactions were factors. For the RNAseq analysis, the differential expression of each gene in the irradiated and control groups was studied by a two-sample *t*-test. The *p*-values were adjusted by an FDR of 0.05 to determine what genes were significantly differentially expressed. For the mouse survival data, Kaplan–Meier survival curves were plotted for each group. Each treated group was compared to the control using the two-sided log-rank test. In these analyses, *p* < 0.05 was regarded as significant. As these were exploratory analyses, we did not adjust *p*-values for multiple comparisons.

## 3. Results

### 3.1. Survival Percentage Depended on the Bacterial Dose Administered for Effective Radiation Mitigation

To highlight the dynamic of radiation mitigation by the genetically engineered probiotic LR-IL-22, we conducted a dose–response experiment using the C57BL/6 mice following 9.25 Gy of total body irradiation (TBI) and bacterial gavages with doses ranging from 10^6^ to 10^10^ cells (Figure 1). The mice received their respective treatments 24 h after irradiation. We noticed that radiation mitigation rates were optimal after gavages of 10^9^ and 10^10^ LR-IL-22 as this correlated with significantly higher percentages of survival compared to the control irradiated group. The differences between the 10^9^ and 10^10^ cells were not statistically significant. 

### 3.2. Increased Overall Survival Was Observed upon Administration of LR-IL-22 but Not IL22 or LR Alone

The overall survival of the C57BL/6 mice was compared between the following four different arms: the control (vehicle) group, the LR group, the IL-22 group, and the LR-IL-22 group following 9.25 Gy TBI. The mice received 10^9^ bacterial cells of LR or its recombinant LR-IL-22 through intra-oral gavages or 0.1 mg/kg of the protein IL-22 via intra-peritoneal (IP) injections. Gavage of the genetically engineered probiotic was noted to significantly improve the overall survival of the mice (Figure 2; *p* = 0.0078). Improved survival was also seen in the mice receiving solely IP injections of the cytokine; however, statistical significance was not observed when compared to the control group (*p* = 0.0677).

### 3.3. Effect of the LR-IL-22 Gavage on Survival in the Male and Female Mice Following Irradiation

It has been reported that sex hormones are a potent driver of differences in the microbiome, and hormones, such as estrogen and progesterone, are radioprotective in nature, and so we decided to evaluate whether the beneficial of effect of LR-IL-22 was gender-specific. Hence, to better understand the effect of LR-IL-22 as a radiation mitigator and assess whether males responded to the treatment equally well, we conducted separate experiments on the male and female C57BL/6NTac mice. The mice (n = 15 per group) were randomized into one of the following four different arms: 9.25 Gy TBI, 9.25 Gy TBI plus LR-IL-22, 13.3 Gy PBI, and 13.3 Gy PBI plus LR-IL-22. The mice received their respective treatments 24 h after total body irradiation (TBI) or partial body irradiation (PBI), where 5% of the bone marrow was preserved. The control irradiated-only mice received a placebo (200 µL saline gavage) and the experimental mice received LR-IL-22 gavages (10^9^ bacteria in 200 µL of PBS). The mice were followed for thirty days, and their overall survival and probability of survival were documented. We observed that both the male and female mice receiving the genetically engineered probiotic had significant survival advantages (Figure 3A; *p* = 0.0078 for the males and *p* = 0.0139 for the females), as well as increased probabilities for survival (Figure 3B; *p* = 0.0063 for the males and *p* = 0.0252 for the females) compared to the same-sex irradiated-only mice; however, the probability of survival rates were not significant when comparing the female mice receiving LR-IL-22 to their male counterparts (*p* = 0.6329). Interestingly, the probability of survival rates were significantly increased between the female irradiated-only mice and the irradiated male mice treated with the genetically engineered probiotic for the PBI-irradiated mice (*p* = 0.0388). 

### 3.4. Differential Gene Expression in Irradiated Stem Cells in Mice Treated with and without TBI and LR-IL-22

We performed RNAseq analyses using the Lgr5+GFP+ mice. The male and female mice were divided into the following four groups (n = 28 male and 28 female mice): (1) 0 Gy (n = 4 males and 4 females), (2) 12 Gy TBI (n = 8 males and 8 females), (3) 12 Gy TBI plus LR (n = 8 males and 8 females), and (4) 12 Gy TBI plus LR-IL-22 (n = 8 males and 8 females). Twenty-four hours after irradiation, the mice were gavaged with either LR or LR-IL-22 (10^9^ bacteria). We observed that stem cell gene expression varied widely following exposure to TBI. Genes were either upregulated or downregulated following irradiation, and many were rescued solely by the genetically engineered probiotic LR-IL-22. In fact, 93% of the genes downregulated by radiation and rescued (upregulated) by LR-IL-22, were also rescued by LR compared to only 32% of the genes upregulated by radiation and rescued (downregulated) by LR-IL-22 that were also rescued by LR (Figure 4A,B). Upon closer look, 135 of the genes were uniquely rescued and, thus, downregulated by LR-IL-22 compared to 43 of the genes that were uniquely rescued and, thus, upregulated by LR-IL-22. In the sections that follow, we provide detailed descriptions of all the pertinent findings. A list of all the differentially expressed (DE) genes is available in the Appendix A.

#### 3.4.1. Genes Downregulated by LR-IL-22

Among the rescued and uniquely downregulated DE genes by LR-IL-22 were many that are involved in promoting inflammation, including ALOX5, which is the rate-limiting enzyme for the biosynthesis of leukotrienes [11], Casp4, which encodes a protein that is a member of the cysteine-aspartic acid protease (caspase) family [12], and OGT, which encodes a glycosyltransferase that catalyzes the addition of a single N-acetylglucosamine in O-glycosidic linkage to serine or threonine residues [13]. Furthermore, many of the genes rescued and, thus, downregulated by LR-IL-22 are involved in promoting tumorigenesis, including MAP2K3, which encodes a protein that belongs to the MAP kinase family of proteins, and LIMD2, which codes for a protein involved in the signal transduction cascade that links integrin-mediated signaling to cell motility and metastatic behavior [14,15]. Similarly, KCTD12, CCDC129, CCNG1, several S100 genes, KRT23, and ST6GalNAc6, which are all involved in tumorigenesis, metastatic behavior, and resistance to radiation [16,17,18,19,20,21,22,23,24,25], were all downregulated solely by LR-IL-22. Also observed was the rescue of many mitochondrial genes, including DDIT4, a stress-responsive protein whose expression is increased under hypoxic conditions and causes cell death through the regulation of mTOR activity and reductions in thioredoxin-1 [23]. We analyzed the GO pathways associated with the downregulated genes that were rescued by LR-IL-22 in the irradiated GFP+ stem cells (Appendix A). The relevant pathways included the positive regulation of wound-healing, acute phase responses, cell-wall disruptions, and nutrient responses, among others.

#### 3.4.2. Genes Upregulated by LR-IL-22

Among the rescued and uniquely upregulated DE genes by LR-IL-22 were many that play a fundamental role in protecting cells against external stressors, including NR2C2, which encodes a protein that belongs to the nuclear hormone receptor family, and MAPK8, which encodes a protein that belong to the MAP kinase family of proteins. Similarly, some of the upregulated genes were those that protect against tumorigenesis and disease progression, such as TSC22D1, which plays a critical role in RNA polymerase II activity [26]. Many were also found to be involved in mitochondrial functioning, including Pisd, which encodes an inner mitochondrial membrane protein.

### 3.5. Intraoral Administration (Gavage) of LR-IL-22 Induced PD-L1 Protein Release in Ovarian Cancer Cells, Mobilized CD8+ T Cells in the WAI Mice, and Improved Overall Survival When Combined with a PD-L1 Inhibitor

Following the IP injection of the 2F8cis cells into the MUC1.Tg WAI (6.0 Gy × 4 daily fractions) mice, treatment was delivered for four consecutive days as of day seven post-IP injection. A subgroup of mice then received intraoral gavages of LR or LR-IL-22 at 24 h and 72 h after the first fraction of WAI. At day nine, the quantification of the PD-L1 expression showed significant increases in the PD-L1-positive cells in the WAI mice (*p* = 0.0225) compared to the control, with further increases when LR-IL-22 was added to the WAI mice (*p* = 0.0058 and *p* = 0.0217 compared to the control and the WAI mice, respectively). These results were further validated by a flow analysis (Figure 5A) and immunohistochemistry of sections of explanted tumor, in which a higher intensity of green fluorescence, representing PD-L1 expression, was noted in the tumor cells of the WAI plus LR-IL-22 mice (Figure 5B). Similarly, significant increases in the numbers of intra-tumoral CD8+ T cells were noted on day nine in the WAI-treated mice, as well as in the WAI plus LR-IL-22-gavaged mice, compared to the control’s untreated tumors (*p* = 0.0126 and *p* = 0.0011, respectively). The increases were much higher in the mice receiving additional intra-oral gavages of LR-IL-22 (*p* = 0.0277). The latter results were further validated by flow analysis and immunohistochemistry of the tumors (Figure 6A,B).

### 3.6. Migration of LR Bacteria into the 2F8cis Ovarian Tumors

Anaerobic bacteria have been shown to migrate and colonize in the hypoxic centers of solid tumors, and this may, in turn, may reduce tumor growth rates. The MUC1.Tg female mice were injected intraperitoneally with 2F8cis tumor cells. Half of the mice were irradiated to 6 Gy × 4 at seven days after tumor injection and sacrificed three days following irradiation. Following intraperitoneal injection of the tumor cells, we confirmed the expression of hypoxia-inducible factor-1α (HIF-1α) that remained stable after irradiation (Figure 7A). After irradiation, we observed the modest upregulation of p16. To investigate whether the LR bacterial strain could migrate into and colonize the ovarian tumors, 2F8cis tumor cells (10^6^ cells) were injected intraperitoneally (n = 4), and one week later, the mice were gavaged daily for three consecutive days with LR (10^9^ bacterial cells). On day four, the mice were sacrificed, and their tumor homogenates were incubated on erythromycin-containing agar plates for the selection of the erythromycin-resistant LR. Forty-eight hours later, LR bacterial colonies from the 2F8cis tumor homogenates were observed to form (Figure 7B). To confirm our findings that the erythromycin-resistant bacterial colonies retrieved from the 2F8cis tumors were, indeed, *Limosilactobacillus reuteri*, we performed 16sRNA gene PCR (Figure 7C), and the PCR products were confirmed by Sanger sequencing. 

## 4. Discussion

The current study successfully illustrated the effectiveness of the second-generation probiotic *Limosilactobacillus reuteri* in releasing IL-22 (LR-IL-22) as an effective radiation mitigator that manifests its role by modifying both gene expression and the cellular microenvironment following irradiation. The intraoral gavage of LR-IL-22 enabled adequate transit of the cytokine through the stomach without being released prior to reaching the intestines. Once in the intestines, IL-22 was released from the genetically engineered probiotic following bacteriophage lysis, a process which we previously described [5]. The success of LR-IL-22 as a mitigator is largely attributed to its mode of delivery. In fact, intraoral gavage enables the cytokine to be entirely released locally in the intestinal crypts, which is not the case when the drug is delivered intravenously. We previously reported that LR-IL-22 treatment maintained intestinal barrier function and modulated the release of irradiation-induced biomarkers of inflammation [5]. We now report the underlying mechanism of mitigation via alteration of the tumor microenvironment and differential gene expression.

Based on our RNAseq data, irradiation was noted to have a significant impact on stem-cell gene expression as it upregulated the expression of many detrimental genes and downregulated that of beneficial ones. Although the expression of some of the genes was rescued by both the LR and LR-IL-22 gavages, many desirable genes were solely rescued by the genetically engineered probiotic LR-IL-22. 

Indeed, radiation upregulated genes for inflammation and tumorigenesis. ALOX5 is crucial for leukotriene biosynthesis and activates downstream genes for chronic inflammation [11]. Casp4 has been implicated in chronic IBD inflammation [12]. OGT, which was upregulated after radiotherapy, inhibits STAT3 signaling in innate immune cells [13]. Radiation also induced many genes that promote tumorigenesis, such as MAP2K3 and RAS, which aid in the oncogenic transformation of primary cells [27]. Likewise, KCTD12 promotes CDC25B/CDK1/Aurora A-dependent G2/M transition. Aurora A is involved in ovarian and breast epithelial carcinomas [17]. CCDC129 has been shown to play a critical role in the invasion, metastasis, and pathogenesis of malignant cells [16]. Furthermore, CCNG1, a target gene of TP53, promotes high-grade serous ovarian cancer tumorigenesis [18]. 

We determined that S100A1, S100A11, and S100A13 were also upregulated by irradiation and downregulated by LR-IL-22. These genes encode proteins that belong to the S100 family of proteins and function in motility, invasion, and tubulin polymerization, therefore playing major roles in tumor metastasis. Specifically, S100A11 plays a central role in promoting the growth, invasion, and migration of ovarian cancer cells. Thus, the knockdown of S100A11 expression suppresses ovarian cancer cell growth and invasion [19,25], and S100A3 expression is associated with chemoresistance of cancer cells [20]. Similarly, the knockdown of KRT23, a member of the keratin family, decreases proliferation and affects the DNA damage response of tumor cells by rendering them more sensitive to ionizing radiation [21]. ST6GalNAc6, a gene that was also rescued and downregulated by the addition of LR-IL-22 following TBI, belongs to a family of sialyltransferases that modify proteins and ceramides on a cell’s surface to alter cell–cell or cell–extracellular matrix interactions. Its stimulation induces the proliferation, migration, and invasion of ovarian cancer stem cells through the Akt signaling pathway [22]. Moreover, the overexpression of PLK2 blocks the cell cycle in the G0/G1 phase, while its downregulation decreases the number of G0/G1 phase cells but increases cell vitality. In epithelial ovarian cancer, CpG island methylation causes PLK2 downregulation, which is related to paclitaxel and platinum tolerance and postoperative recurrence ( ). Finally, the radiation-induced upregulation of DDIT4, by acting as a negative regulator of mTOR, induces anti-tumor therapy resistance. Additionally, DDIT4 is associated with the cell invasion and migration of ovarian epithelial cells, facilitating the expression of autophagy-related proteins and inhibiting tumor cell apoptosis [23]. All of the above-described genes were solely rescued and downregulated by LR-IL-22, thus limiting radiation-induced inflammatory processes, evasion by the tumor cells, and disruption of mitochondrial activity. LR, on the other hand, had no effect on any of these processes. 

Radiation induced the downregulation of many genes that protect cells from external stress, including NR2C2, which plays a critical role in protecting cells from radiation-induced oxidative stress and damage [28]. Similarly, MAPK8, important in cellular differentiation, proliferation, and development, is downregulated by radiation [29], and it was upregulated by LR-IL-22, an effect which was not noticeable with the intraoral gavage of LR, further highlighting the role of LR-IL-22 as a radiation mitigator.

The cellular response to hypoxia includes the expression of HIF-1α [30], and there is a correlation between the extent of tumor hypoxia and overall disease prognosis [31]. We also observed HIF-1α (hypoxia inducible factor-1 alpha) expression in the intraperitoneal tumors, together with the modest upregulation of p16 positive cells, after radiation.

Several clinical studies have highlighted the fact that radiotherapy could potentially be used as monotherapy in oligometastatic disease with the advent of IMRT and SBRT; however, intestinal toxicity has remained problematic [4,32,33,34]. Herein, we reported a novel approach to intestinal radioprotection during fractionated WAI for widespread abdominal malignancies, including ovarian cancer. The MUC1.Tg mouse model we used is an established replica which mimics human ovarian cancer. At baseline, the CD8+ T cells infiltrated the 2F8cis tumors, giving them an immunological ‘‘hot’’ phenotype and positively correlating with PD-L1 expression. Recently, it has been shown that combining these treatment modalities has a synergistic effect on ovarian cancer management as the application of radiotherapy induces the in situ vaccination of tumor cells and apoptosis of the T-reg lymphocytes, which could further enhance cell-mediated immune responses and cytotoxic T cell activity [35,36,37,38,39,40,41,42,43]. Fractionated WAI further upregulated CD8+ T cell infiltration and PD-L1 cellular expression, which may theoretically increase the effectiveness of immunotherapies. This forms the basis for using WAI with targeted therapies, including immune checkpoint inhibitors in advanced or recurrent epithelial ovarian cancer.

## 5. Conclusions

The present data support the use of LR-IL-22 in clinical protocols as a mitigator of radiotherapy in women suffering from advanced, recurrent, or cisplatin-resistant ovarian cancer with peritoneal disease dissemination who may benefit from WAI. The addition of LR-IL-22 to a combined treatment modality with fractionated WAI and systemic chemotherapy and immunotherapy regimens will facilitate a safe and effective protocol for reducing tumor burden, increasing survival, and improving quality of life in patients with widespread abdominal ovarian cancer.

## Figures and Tables

**Figure 1 cancers-16-00474-f001:**
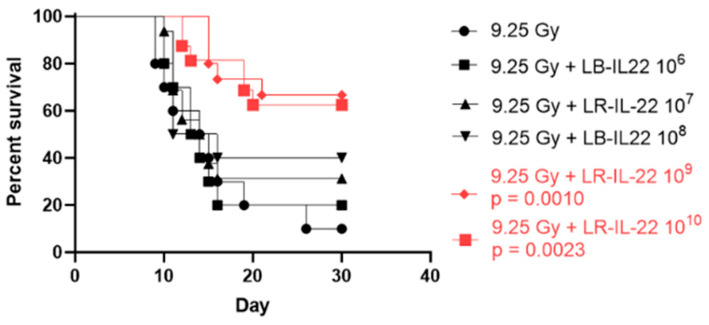
Survival of the C57BL/6 mice gavaged 24 h after 9.25 Gy TBI with different numbers of LR-IL-22. Female mice (n = 90) were treated with increasing doses of LR-IL-22 (n = 15 mice per group). Significant increases in survival compared to the control-irradiated arm were only seen after gavages of 10^9^ and 10^10^ cells (*p* = 0.0010 for the 10^9^ bacteria and *p* = 0.0023 for the 10^10^ bacteria).

**Figure 2 cancers-16-00474-f002:**
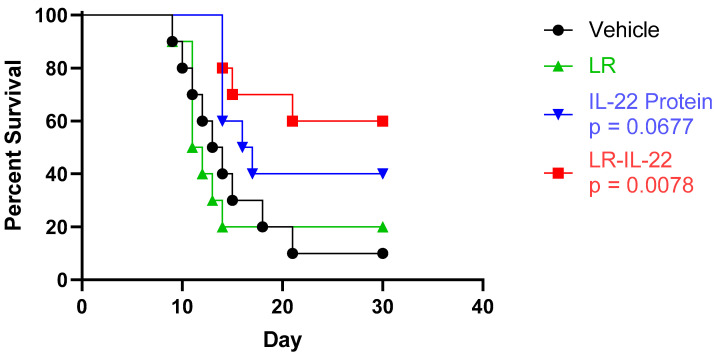
Survival of the C57BL/6 mice after treatment with LR, IL-22, and LR-IL-22. A total of 40 female mice were divided into the following four different groups (n = 10 per group): the vehicle (control) group, the LR group, the IL-22 group, and the LR-IL-22 group, and they were irradiated to 9.25 Gy. Twenty-four hours after irradiation, the mice were gavaged with 10^9^ LR or LR-IL-22 bacteria or IP-injected with the IL-22 protein (0.1 mg/kg). They were then observed for 30 days to assess their overall survival. The LR and LR-IL-22 were administered through intraoral gavage, and the IL-22 was given through intraperitoneal injections.

**Figure 3 cancers-16-00474-f003:**
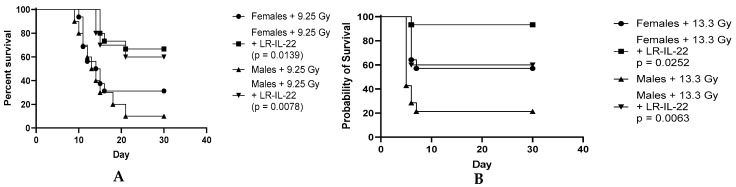
Increased overall survival rates and survival probability rates for the irradiated male and female C57BL/6 mice after LR-IL-22 gavage. Male and female C57BL/6 mice (n = 120) were randomized after irradiation into the following four different arms following sex stratification (n = 15 per group): 9.25 Gy TBI, 9.25 Gy TBI plus LR-IL-22, 13.3 Gy PBI, and 13.3 Gy PBI plus LR-IL-22. The mice treated with LR-IL-22 (10^9^ bacteria) received it through intraoral gavages 24 h following irradiation. During PBI, 5% of the bone marrow was shielded. (**A**) Overall survival after TBI. (**B**) Probability of survival after PBI. TBI, total body irradiation; PBI, partial body irradiation.

**Figure 4 cancers-16-00474-f004:**
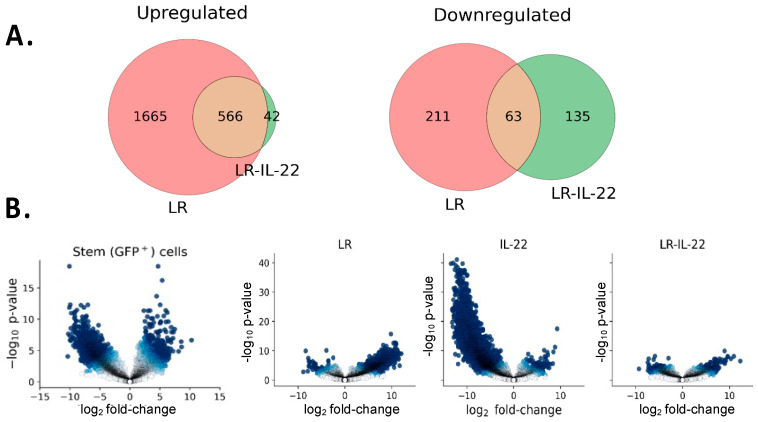
Differential gene expression following 12 Gy TBI plus treatment with LR, IL-22, or LR-IL-22. Mice were treated with 12 Gy TBI and received saline in irradiation only, 0.1 mg/kg of IL-22 protein as an IP injection, LR or LR-IL-22 via intra-oral gavage 24 hours later. (**A**) balloon plot describing shared and unique rescue of genes by LR and LR-IL-22 (**B**) Volcano plots for stem cells illustrating differential gene expression based on the treatment received.

**Figure 5 cancers-16-00474-f005:**
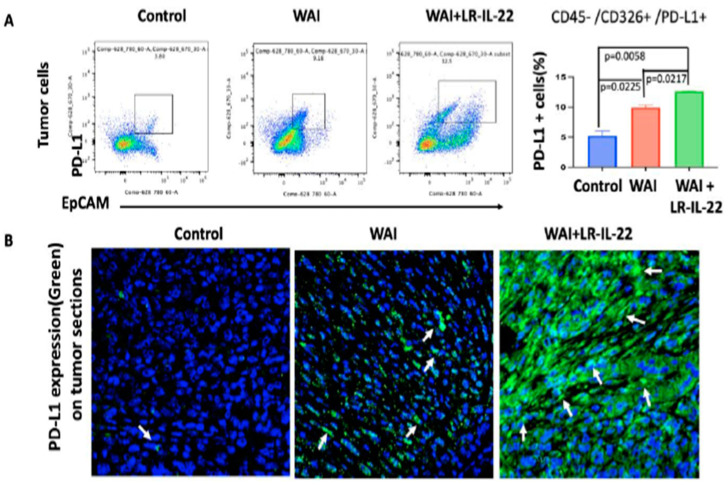
LR-IL-22 gavage- and WAI-induced PD-L1 expression in the 2F8cis tumor cells in the MUC-1 mice. (**A**) A flow cytometric analysis of the CD45−/CD326+/PD-L1+ 2F8cis tumor cells was performed using freshly isolated intraperitoneal 2F8cis tumors from the MUC-1 mice. The mice were given four fractions of whole abdomen irradiation (WAI) (6 Gy × 4) for four consecutive days alone or together with LR-IL-22 gavages on day two and day four (first radiation dose on day one). Tumors were collected on day nine, and the single-cell suspensions were analyzed by flow cytometry followed by staining. We used EpCAM to distinguish the PD-L1-expressing non-tumor cells from the 2F8cis tumor cells. The numbers of PD-L1-expressing 2F8cis tumor cells were significantly higher in the WAI and WAI plus LR-IL-22 groups compared to the no-radiation tumor-only group. The WAI plus LR-IL-22 treatments further increased the numbers of PD-L1-expressing tumor cells compared to the WAI-only treatments (n = 2–3 mice, 5–7 tumor nodules per mouse; *p* values were calculated by *t*-tests). The number of mice required is discussed in the power and sample size determination section. (**B**) Representative immunofluorescent images of the excised 2F8cis tumor tissue sections on day nine showing the PD-L1 expression (arrows) in the tumor cells of the WAI-treated and WAI plus LRIL-22-treated groups compared to tumor-only control group. PD-L1 expression levels were further increased in the WAI plus LRIL-22 group compared to the WAI-only group (n = 3). The green fluorescence represents PD-L1 expression and the blue (DAPI) staining represents cell nuclei.

**Figure 6 cancers-16-00474-f006:**
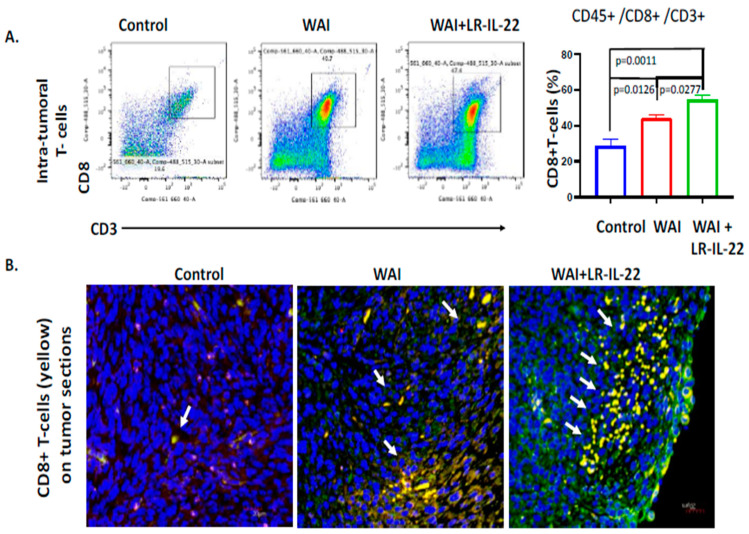
LR-IL-22 gavages and WAI treatment increased the intra-tumoral CD8+ T cells in the 2F8cis tumors in the MUC-1 mice. (**A**) Flow cytometric analysis of the tumor-infiltrating CD45+/CD3+/CD8+ T cells was performed using freshly isolated intraperitoneal 2F8cis tumors from the MUC-1 mice. The mice were given four fractions of whole abdomen irradiation (WAI) (6 Gy × 4) for four consecutive days, and they were given LR-IL-22 by gavage on day two and day four (first radiation dose on day one). Tumors were collected on day nine, and the single-cell suspensions were analyzed. The CD45+/CD3+/CD8+ T cells were quantified using flow cytometry. There were significant increases in the tumor-infiltrating CD8+ T cells in both the WAI-only and WAI plus LR-IL-22 groups compared to the tumor-only control. The WAI plus LR-IL-22 treatment further increased the numbers of tumor-infiltrating CD8+ T cells compared to the WAI-only group (n = 2–3 mice, 5–7 tumors per mouse; *p* values were calculated by *t*-tests). (**B**) Representative immunofluorescent images of the 2F8cis excised tumor tissue sections on day nine showing the increased abundances of CD45+ and CD8+ cells (arrows) in the tumors from the WAI-treated and WAI plus LR-IL-22-treated groups compared to the tumor-only control group. The prevalence of intra-tumoral CD45+ and CD8+ cells was further increased in the WAI plus LR-IL-22 group compared to the WAI-only group (n = 3). The yellow fluorescence represents the co-expression of CD45 (red) and CD8 (green), and the blue (DAPI) staining represents cell nuclei.

**Figure 7 cancers-16-00474-f007:**
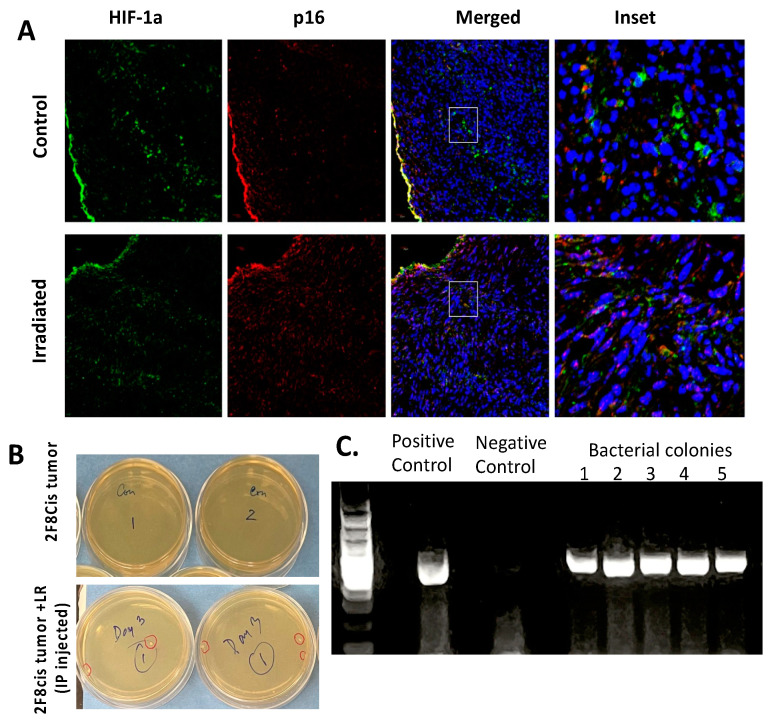
Upregulation of p16 and downregulation of HIF-1α in the irradiated tumors. Muc1.Tg mice (n =10) received intraperitoneal injections of 2F8Cis tumor cells (1 × 10^9^ cells). After seven days, half of the mice (n = 5) were irradiated to 6 Gy × 4 (WAI). Three days later, tumors isolated from the control and irradiated mice (**A**) were stained for HIF-1α and P16. (**B**) The LR bacterial colonies from the 2F8cis tumor homogenates grown on erythromycin-resistant plates at 48 h. (**C**) The 16s RNA PCR products from the bacterial colonies.

## Data Availability

The data that support the findings of this study are available upon reasonable request.

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
