# Peer review of "Genetically Engineered Probiotic Limosilactobacillus reuteri Releasing IL-22 (LR-IL-22) Modifies the Tumor Microenvironment, Enabling Irradiation in Ovarian Cancer"

_cancers, 2024, doi:10.3390/cancers16030474_

Round 1

Reviewer 1 Report

Comments and Suggestions for Authors

thank you for this interesting paper. 

A few comments:

Line 36 and 385 can you really say "will" rather than may or could as there are no human studies yet

line 174 needs completion

line 186 delete the "of" before effect

If this is an effective radiation toxicity modulator we need to know about toxicities> do LR and LR-1Il-22 have any toxic effects?

Author Response

We want to thank Reviewer 1 for his comments on the quality of the manuscript.  Listed below are our responses for your suggestions to improve our manuscript.

  1. For lines 36 and 385 (now 415) we have changed will to may since we have not begun human studies.
  2. We have completed the sentence on line 174 (now 175).The sentence begins with improved survival…
  3. We have removed the work of on line 186 (now 191).
  4. We have added a paragraph discussing the lack of toxicity resulting from the gavage of LR or LR-IL-22.The paragraph is in the discussion section at line 339.

Reviewer 2 Report

Comments and Suggestions for Authors

I read with great interest the manuscript, which falls within the aim of this Journal and offers a high-quality overview of the topic.
Methodology is accurate and conclusions are supported by the data analysis. The tables and figures are clear and interesting.
Although the manuscript can be considered already of high quality, I would suggest taking into account the following minor recommendations:

- I suggest another round of language revision, in order to correct a few typos and improve readability.

- I suggest reading and adding recent evidence about the importance of adequate management in patients with ovarian cancer, to improve prognosis and the possibility of tailored management. I suggest authors to read and insert in references the following article: Tullio Golia
D'Augè, Andrea Giannini,  Giorgio Bogani,  Camilla Di Dio,  Antonio Simone Laganà,  Violante Di Donato,  Maria Giovanna Salerno,  Donatella Caserta,  Vito Chiantera,  Enrico Vizza,  Ludovico Muzii,  Ottavia D’Oria. Prevention, Screening, Treatment and Follow-Up of Gynecological Cancers: State of Art and Future Perspectives. Clin. Exp. Obstet. Gynecol. 2023, 50(8), 160. https://doi.org/10.31083/j.ceog5008160.

-What is already known on this subject?

-What do the results of this study add? 

-What are the implications of these findings for clinical practice and/or further research? It is important to report the results obtained by the authors in the context of clinical practice and to adequately highlight what contribution this study adds to the literature already existing on the topic and to future study perspectives.

- Considering the topic and results of this study, I suggest that authors evaluate and cite relevant evidence about therapeutic strategies for advanced ovarian cancer. I would be glad if the authors discuss this important point.

Comments on the Quality of English Language

  I suggest a round of language revision, in order to correct few typos and improve readability.

Author Response

We want to thank Reviewer 2 for the comments on the manuscript being of high quality.  We also want to thank the reviewer for suggestions which will make the manuscript better.

  1. We have gone over the  and corrected the typos and improved the readability of the manuscript.
  2. We have added new references including the one you suggested to better explain why new methods are needed to treat ovarian cancer.We have also added how the use of LR-IL-22 may be beneficial in the treatment of ovarian cancer.